# Geographical Detection of Traffic Accidents Spatial Stratified Heterogeneity and Influence Factors

**DOI:** 10.3390/ijerph17020572

**Published:** 2020-01-16

**Authors:** Yuhuan Zhang, Huapu Lu, Wencong Qu

**Affiliations:** 1Institute of Transportation Engineering and Geomatics, Tsinghua University, Beijing 100084, China; yh-z15@mails.tsinghua.edu.cn; 2Road Transport Books Center, China Communications Press Co., Ltd., Beijing 100011, China; quwencong@163.com

**Keywords:** spatial analysis, spatial statistics, geographical detectors, stratified heterogeneity, factors, traffic accident, nonlinear interaction

## Abstract

The purpose of this paper is to investigate the existence of stratification heterogeneity in traffic accidents in Shenzhen, what factors influence the casualties, and the interaction of those factors. Geographical detection methods are used for the analysis of traffic accidents in Shenzhen. Results show that spatial stratification heterogeneity does exist, and the influencing factors of fatalities and injuries are different. The traffic accident causes and types of primary responsible party have a strong impact on fatalities and injuries, followed by zones and time interval. However, road factors, lighting, topography, etc., only have a certain impact on fatalities. Drunk driving, speeding over 50%, and overloading are more likely to cause more casualties than other illegal behaviors. Speeding over 50% and speeding below 50% have significant different influences on fatalities, while the influences on injuries are not obvious, and so do drunk driving (Blood Alcohol Concentration ≥ 0.08) and driving under the influence of alcohol (0.08 > Blood Alcohol Concentration ≥ 0.02). Both pedestrians and cyclists violating the traffic law are vulnerable to fatality. Heavy truck overloading is more likely to cause major traffic accidents than minibuses. More importantly, there are nonlinear enhanced interactions between the influencing factors, the combination of previous non-significant factors and other factors can have a significant impact on the traffic accident casualties. The findings could be helpful for making differentiated prevention and control measures for traffic accidents in Shenzhen and the method selection of subsequent research.

## 1. Introduction

Traffic accidents have an important impact on life safety and economic development. A total of 244,937 road traffic accidents occurred in 2018 in China, which caused 63,194 fatalities, 258,532 injuries, and direct economic loss valued at 1.38 billion yuan [1]. Shenzhen is an international innovative metropolis with total urbanization [2] and the highest vehicle density in China [3]. According to the national strategy of China, by 2025, Shenzhen will be built into one of the leading cities in the world in terms of economic strength and quality of development [4]. In the past five years, the fatality rate of 10,000 vehicles in Shenzhen has been declining continuously, reaching 0.95 in 2018, but compared with the developed countries in the world, there is still a gap [3]. In order to narrow this gap, the Traffic Management Research Institute of the Ministry of Public Security launched a pilot data-mining activity of traffic accidents in Shenzhen, which have provided traffic accident data of Shenzhen, and the findings will be used to determine road traffic safety management measures.

Traffic accidents are geographical events. Spatial autocorrelation and spatial stratified heterogeneity are two major features of geographical phenomena [5]. Previous studies have confirmed that traffic accidents also have these two properties [6]. Based on the investigation of the spatial characteristics of traffic accidents in Shenzhen and from the perspective of spatial stratification heterogeneity, this paper compares the effects of various factors on fatalities and injuries and the sub-stratum differences of factors, and studies the types and intensities of the interaction among the factors. The findings could be helpful for making differentiated prevention and control measures of traffic accidents in Shenzhen and selecting methods of subsequent research.

## 2. Literature Review

### 2.1. Influence Factors on Traffic Accidents

Previous studies have revealed many factors affecting traffic accidents’ severity, such as human factors, vehicle conditions, traffic characteristics, road infrastructure, and environmental conditions [7,8]. Aberrant driving behaviors were found to be the most important factor of traffic safety [9]. Parker [10] discussed a three-fold typology of aberrant driving behaviors, namely lapses, errors, and violations, and their threat to the safety of others. Among these, speeding was found to be the main cause of traffic accidents [11]. Wang et al. [12] reviewed the road-related factors affecting road traffic accidents, and concluded that speed, congestion, and road horizontal curvature have mixed effects on road safety and need further examination. Researches have also suggested that vehicle type, road type, time of day, streetlight condition, and weather conditions are important factors that affect the severity of traffic accidents [13]. The occurrence and outcome of traffic crashes have long been recognized as complex events involving interactions between many factors [14]. However, the type and direction of the interactions between factors are rarely studied in depth.

Over the years, generalized linear models (GLM) have been widely used in traffic accident risk factor detection, such as the linear and multiple regressions model, the binary logit and binary probit models, multiple logistic regression, the ordered logit model, and so on [15]. All GLMs suffer from a common underlying limitation that each observation (e.g., a crash or a vehicle involvement) is independent. In reality, this “independence” assumption may often not hold true. When observations are dependent, the actual number of independent samples is less than that of observation samples, the confidence intervals will be wrongly estimated, and hence, the regression coefficients might be biased and the accuracy might be overestimated [16,17]. Furthermore, spatial autocorrelation means that the value of a point is affected by its neighbors, and the closer they are, the more similar (positive autocorrelation) or less similar (negative autocorrelation) they are. This might also lead to biased estimation and even reverse interpretation of influencing factors [18]. Considering these, some spatial analysis models were developed. By capturing the spatial autocorrelation and heterogeneity, the geographically weighted regression (GWR) and the Bayesian regression models were proven to outperform the GLM [19,20]. The models based on linear equations assume that risk factors influence crash frequency/severity in a linear manner. However, researchers have found that non-linear relationships exist between crashes and risk factors [21]. Thus, machine learning methods with little or no prior hypothesis for input variables were introduced to identify risk factors. Examples include artificial neural networks (ANN) [22], boosted regression trees (BRT) [23], support vector machines (SVM) [24], and stacking of several machine learning methods [25]. The major disadvantage of machine learning methods is that they often lack a direct and clear interpretation between accident severity and related variables.

### 2.2. Spatial Stratified Heterogeneity Detection of Traffic Accidents

Spatial stratified heterogeneity refers to uneven distributions of traits, events, or their relationship across a region or, simply, spatial variation of attributes [5]. In statistics, their main performance is that the variance within strata is less than that between strata, such as geographical division, climatic zone, land use map, urban–rural difference, and main functional area [26]. Spatial stratification based on prior knowledge is a feasible method, stratification of heterogeneity recognized by humans, however, may be inconsistent with the true stratified heterogeneity in nature due to the limitations of human intelligence [27]. Many clustering and classification algorithms have been used for segmentation or group traffic accidents. Examples include, k-means [28,29], latent class clustering (LCC) [30], and SVM [31]. However, these methods perform like a ‘black box’ approach and it is difficult to explain the stratification outcome. Although the degree of stratified heterogeneity of a traffic accident’s attribute is an important indicator, few statistical tests for the significance of the degree of spatial stratified heterogeneity are available yet. It is necessary to find a method for judging whether the spatial stratified heterogeneity exits and whether a spatial partition is optimal.

## 3. Data

The traffic accidents data of Shenzhen, China, during 2014–2016 were collected. They were provided by the pilot data-mining activity of traffic accidents in Shenzhen carried out by the Traffic Management Science Research Institute of the Ministry of Public Security of the People’s Republic of China. Based on preliminary analysis of the data, a set of influence factors on injury severity of traffic accidents in Shenzhen is set up. Y indicates the injury severity of traffic accident, measured by injuries and fatalities, respectively. $X_{i}$ indicates the independent variable that may have a significant impact on the injury severity of traffic accidents. The set of influence factors includes five aspects:

A.Geographical regions, including zones $X_{1}$.B.Time of occurrence, including seasons $X_{2}$, day of the week $X_{3}$, and time intervals $X_{4}$.C.Road factors, including road type $X_{5}$, road line type $X_{6}$, road section type $X_{7}$, pavement material $X_{8}$, pavement condition $X_{9}$, and roadside protection type $X_{10}$.D.Management status, including the traffic sign $X_{11}$ and lighting condition $X_{12}$.E.Environment condition, including weather $X_{13}$ and topography $X_{14}$.F.Traffic violation, including primary cause $X_{15}$, whether illegal $X_{16}$, and types of primary responsible party $X_{17}$.

According to the description of the location in the accident record, we got the latitude and longitude of the sites by geocoding with the aid of the application programming interface (API) provided by Tencent (https://lbs.qq.com), which provides location services with high data quality and complete address data. Accident records with vague locations were removed, as those data points could not be used to determine the precise location. Simple traffic accidents and abnormal data were removed too. Finally, 3250 data points were selected. The dataset is divided into two groups, group 1 is accidents with casualties and group 2 is accidents with injuries only. Fatalities (Mean = 0.48, standard deviation (SD) = 0.301) and injuries (Mean = 1.41, SD = 1.055) respectively, indicated the severity of the two groups. The definitions and descriptive statistics of Y and $X_{i}$ are shown in Table 1.

The Moran I test [32] shows that there is a less than 1% likelihood that the pattern of traffic accident fatalities or injuries could be the result of random chance (Moran’s I > 0, Z-value > 2.58 and *p*-value < 0.01). There is spatial autocorrelation in traffic accidents of Shenzhen city. This is one of the important bases for model selection. Ignorance of spatial dependency will lead to biased estimates.

## 4. Methods

If an independent variable has an effect on a dependent variable, the spatial distribution of the independent variable should be consistent with that of the dependent variable. Based on this idea, Wang and Hu [33] proposed a spatial stratified heterogeneity measurement method. Then, it was improved gradually and a systematic statistical method to detect spatial stratified heterogeneity and reveal the driving force behind it was constructed [34,35]. It has a straight physical meaning, and can detect the real interaction between two variables, not limited to multiplicative interactions. Meanwhile, it is immune to multicollinearity and has no linear hypothesis for variables. It has four geographical detectors and has a common name, Geodetector.

### 4.1. Basic Principles

Conceptually, a stratification of heterogeneity is a partition of a study area, where observations are homogeneous within each stratum but not between strata [34]. The q statistic is the foundation of a Geodetector. It is a statistical classification algorithm to find a study area that can minimize the within-strata difference and maximize the between-strata differences. The differences are measured by q value, a ratio of the between- and the within-strata variances. The larger the q value is, the greater the heterogeneity of this study area is. Meanwhile, if the differences are caused by an independent variable and its classification, it means that this independent variable has an influence on the dependent variable. The greater the q value is, the greater the influence is.

### 4.2. Factor Detector

Are there some geographical strata responsible for an observed spatial pattern? This can be detected by a factor detector. Using the q value to measure how many differences there are in the Y spatial distribution can be explained by an independent variable X. The value of q is:(1)$$q=1-\frac{\sum_{h=1}^{H}N_{h}\sigma_{h}^{2}}{N\sigma^{2}}=1-\frac{SSW}{SST}$$

Note that h=1,2,…,H is the stratification (classification) of the dependent variable Y or the independent variable X, $N_{h}$ is the unit number of sub-stratum h, N is the unit number of the whole strata, and the $\sigma_{h}^{2}$ and $\sigma^{2}$ are the variances of the variables Y in the H sub-stratum and the whole strata, respectively. SSW denotes within-strata sum of variances and SST denotes between-strata sum of variances. The value of q is between 0 and 1. The larger the q value of the dependent variable Y is, the more obvious the spatial differentiation is. The q value of the independent variable X explains 100⋅q%
Y. The larger the value, the more consistent the spatial distribution of X and Y, and the stronger the interpretation ability of X to Y. In extreme cases, if q=1, then the independent variable X completely controls the dependent variable Y. If q=0, then the independent variable X has nothing to do with the dependent variable Y.

### 4.3. Influence Detector

The influence detector is used in the search for strata of potential security hazards. It compares the difference of average values between sub-strata to find the aggregation or fusion strata (regions). The null hypothesis is: there is no significant difference between the mean attributes of sub-stratum 1 and sub-stratum 2. The significance is tested by t statistics:(2)$$t_{{\bar{X}}_{h=1}-{\bar{X}}_{h=2}}=\frac{{\bar{X}}_{h=1}-{\bar{X}}_{h=2}}{{\left[\frac{\mathrm{var}({\bar{X}}_{h=1})}{n_{h=1}}-\frac{\mathrm{var}({\bar{X}}_{h=2})}{n_{h=2}}\right]}^{1/2}}$$
where, $\bar{X_{h}}$ is the mean value of attributes in the *H*-stratum of the independent variable X, $n_{h}$ is the number of sample units in the *H*-stratum of the independent variable X, and var() is the variance. The statistic t approximately obeys the Student’s *t* distribution.

### 4.4. Ecological Detector

The main function of the ecological detector is to compare the influence of two independent variables, $X_{1}$ and $X_{2}$, on the spatial distribution of the dependent variable Y. It is measured by F statistics:(3)$$\begin{array}{l} F=\frac{N_{X1}(N_{X2}-1)SSW_{X1}}{N_{X2}(N_{X1}-1)SSW_{X2}}\\ SSW_{X1}=\sum_{h=1}^{H_{1}}N_{h}\sigma_{h}^{2},SSW_{X2}=\sum_{h=1}^{H_{2}}N_{h}\sigma_{h}^{2} \end{array}$$
where, $N_{X1}$ and $N_{X2}$ are the sample sizes of the independent variables $X_{1}$ and $X_{2}$ respectively, and $SSW_{X1}$ and $SSW_{X2}$ are the sum of intra-stratum variances of the independent variables $X_{1}$ and $X_{2}$, respectively.

### 4.5. Interaction Detector

Different independent variables can interact with each other. The interaction detector is designed to evaluate whether the interaction of two independent variables, $X_{1}$ and $X_{2}$, will enhance the contribution to the dependent variable Y, and then discriminate the type of interaction.

The method is as follows: first, calculate the q values of $X_{1}$ and $X_{2}$, q(Y/X1) and q(Y/X2) (abbreviated as q(X1), q(X2)), and then calculate the q values of X1∩X2 (means superposition of X1 and X2) q(Y/X1∩X2) (abbreviated as q(X1∩X2)), and finally, compare q(X1), q(X2) and q(X1∩X2). The interaction types and strengths are shown in Table 2. The approach is feasibly extendable to three or more independent variables.

## 5. Experimental Results

### 5.1. Spatial Stratified Heterogeneity and Influence Factors

Factor detection reveals spatial stratified heterogeneity and influence factors of fatalities and injuries in Shenzhen traffic accidents (Table 3). The q value represents the explanatory power of factors to fatalities or injuries. The factor with a q value greater than 0 and passing the significance test is the factor with spatial stratification heterogeneity, and also the factor with a significant influence. Therefore, there is spatial stratified heterogeneity in both fatalities and injuries at the 95% confidence level. The biggest q value of fatalities and injuries is 0.094 and 0.12 respectively, and the corresponding influencing factors are all primary cause. That is to say, primary cause can explain the spatial stratification heterogeneity of 9.4% of fatalities and 12% of injuries.

### 5.2. Sub-Strata Comparison of Influencing Factors

Risk detection shows that there are significant differences in the mean value of fatalities and injuries between sub-strata of each influencing factor. The confidence level below is 95 percent.

#### 5.2.1. Primary Cause

Figure 1 shows the sub-strata comparison result of fatalities and injuries causes. For fatalities, the sub-strata are 3, 16, 6, 5, 17, 13, 1, 20, 18, and 12 in descending order of mean value, but only sub-strata 3, 16, 6, 5, 17, 13, 1, and 12 were significantly higher than that of other sub-strata, except 18 and 20. Amazingly, the mean value of injuries in sub-stratum 3 and 5 exceeded 4.0, followed by sub-stratum 4, 10, 18, 2, and 1. The mean value of injuries in sub-stratum 1, 2, 3, 5, 9, 10, 11, 19, and 21 was significantly higher than that in sub-stratum 12, 13, 16, 17, and 20.

It is noteworthy that there are differences in the heterogeneity of primary cause between fatalities and injuries. The mean value of fatalities and injuries in sub-stratum 1, 3, and 5 was higher, but the mean value of fatalities in sub-stratum 12, 13, 16, and 17 were significantly higher, while the mean value of injuries were significantly lower. For the mean value of fatalities, sub-stratum 1 was significantly higher than sub-stratum 2, and sub-stratum 3 was also significantly higher than sub-stratum 4. For the mean value of injuries, however, the difference between sub-stratum 1 and sub-stratum 2, and sub-stratum 3 and sub-stratum 4, were not significant. No matter for fatalities or injuries, sub-stratum 3 was significantly higher than sub-stratum 1. That is to say, compared with all types of primary cause, the mean values of fatalities and injuries caused by drunk driving, speeding over 50%, and overloading were all higher. However, the mean values of fatalities caused by illegal road occupying, illegal backing, helmet violation, and illegal entering onto highway were significantly higher, while the mean values of injuries were significantly lower. For the mean value of fatalities, drunk driving was significantly higher than driving under the influence of alcohol and speeding over 50% was significantly higher than speeding below 50%. For the mean of injuries, however, the difference between drunk driving and driving under the influence of alcohol, and speeding over 50% and speeding below 50%, were not significant. No matter for fatalities or injuries, speeding over 50% was significantly higher than drunk driving.

#### 5.2.2. Types of Primary Responsible Party

Figure 2 shows the sub-strata comparison result of fatalities and injuries of types of primary responsible party.

For fatalities, sub-stratum 1, sub-stratum 6, sub-stratum 5, and sub-stratum 4 had the highest mean fatalities. There was no significant difference in the mean values of the first three sub-strata and it was significantly higher than that of sub-stratum 2, and the mean value of sub-stratum 2 was significantly higher than that of sub-stratum 3, 7, 8, and 10. Although the mean value of sub-stratum 4 ranks fourth, it was only significantly higher than that of sub-stratum 8 due to its large variance, while sub-stratum 8 was significantly lower than other layers.

For injuries, it is surprising that the mean values of sub-stratum 4 and 6 were over 3.0, followed by sub-stratum 5 and 7, and were significantly higher than other sub-stratums. Sub-stratum 5 showed no significant difference with others, and sub-stratum 6 was only significantly higher than sub-stratum 1 and 9. Sub-stratum 9 and 1 had the lowest mean value, which was significantly lower than other sub-stratums, except 5 and 8.

It should be noted that there were similarities and differences in the stratification heterogeneity between fatalities and injuries. The mean values of fatalities and injuries in sub-stratum 4, 5, and 6 were higher, but fatalities in sub-stratum 4 and injuries in sub-stratum 5 and 6 were discrete. The differences between the mean values of fatalities and the mean values of injuries in sub-stratum 1, 2, 3, and 7 were significant. According to the mean values of fatalities from high to low, the ranking is sub-stratum 1, 2, 3, 7, while according to the mean values of injuries, the ranking is sub-stratum 7, 2, 3, 1. The mean value of fatalities in sub-stratum 1 was significantly higher, while the mean value of injuries was significantly lower. The mean value of fatalities in sub-stratum 7 was significantly lower, while the mean value of injuries was significantly higher. No matter for the mean value of fatalities or injuries, sub-stratum 2 was greater than sub-stratum 3. That is to say, compared with all types of primary cause, the mean values of fatalities and injuries caused by large and medium buses, light trucks and heavy trucks, are all higher. The differences between the mean values of fatalities and the mean values of injuries caused by pedestrians, non-motorized vehicles, minibuses, and motorcycles were significant. According to the mean values of fatalities from high to low, the ranking of types of primary responsible party was pedestrians, non-motorized vehicles, minibuses, and motorcycles, while according to the mean values of injuries, the ranking was motorcycles, non-motorized vehicles, minibuses, and pedestrians. The mean value of fatalities caused by pedestrians was significantly higher, while the mean value of injuries was significantly lower. The mean value of fatalities caused by motorcycles was significantly lower, while the mean value of injuries was significantly higher. No matter for the mean value of fatalities or injuries caused by non-motorized vehicles, it was greater than that caused by minibuses.

#### 5.2.3. Other Factors

The stratified heterogeneity of geographical zones, road factors, management status, and environment condition affecting fatalities and injuries was examined. The main findings are shown in Table 4.

### 5.3. Interaction of Influencing Factors

The interaction detector was used to check whether two influencing factors work independently or not, and further discriminate the type of interaction. It is notable that there was no pair of factors found to be independent or linear. All the factors were found to enhance each other to increase fatalities and injuries. When any two factors work together, the explanatory power of combination for fatalities and injuries is greater than that of a single factor. Only two types of interaction were involved: nonlinear enhancement and bi-enhancement. For fatalities, the two types account for 93.6% and 6.4% respectively, for injuries, the two types account for 94.5% and 5.5%, respectively.

Ranked by explanatory power, the top 10 interaction details of fatalities and injuries are shown in Table 5 and Table 6, respectively. It is worthwhile to note that the most powerful explanation for both fatalities and injuries comes from the interaction of primary cause and other factors. Especially for fatalities, the top 10 powerful explanations are all so. Another notable result is that some top 10 explanatory power comes from the interaction of a non-significant influencing factor and another factor. For example, day of the week was a non-significant influencing factor for fatalities, but when it works together with primary cause, the combined factor becomes significant and enhances each other to increase fatalities nonlinearly. Seasons was a non-significant influencing factor for the injuries, but after it was combined with primary cause, the combined factor became a significant factor, and the explanatory power surprisingly became 42.1%, ranking first in all combined factors.

Which primary responsible parties, and when they have what kind of fault, will lead to a large number of casualties? The combined factor of primary cause and types of primary responsible party was geographically detected again. The top 10 high-risk lethal behaviors and high-risk injury behaviors with significantly higher mean values of fatalities and injuries are shown in Table 7.

## 6. Discussion

### 6.1. Spatial Stratification Heterogeneity

Through geographical detection, it was found that traffic accidents in Shenzhen have spatial stratified heterogeneity. The biggest difference among sub-stratum was primary cause, followed by types of primary responsible party. Many previous studies have found the stratified heterogeneity in traffic accidents [36,37]. The existence of spatial stratified heterogeneity makes the global model or parameters unable to accurately capture the local characteristics [27]. Therefore, it has an important impact on model selection. Thus, it is suggested that in the analysis and prediction of traffic accident-related factors, spatial stratified heterogeneity detection should firstly be carried out. When there is spatial stratified heterogeneity, these methods including modeling data hierarchically, introducing local variables, or using variable functions, which may perform better. For example, Li et al. [38] compared the performance of geographically weighted Poisson regression (GWPR) and the traditional GLM model, and found that as GWPR can capture the spatial non-stationary relationship between traffic accidents and prediction factors, its prediction performance is better than the GLM model. De Ona et al. [39] first applied latent class cluster (LCC) to split data, then used Bayesian networks (BNs) to model the cut dataset and the complete dataset separately, and found that data-splitting models can identify more factors than models that do not split data. Based on the prior knowledge, Sun et al. [8] established the binary logistic regression analysis model and the decision tree model according to the urban functional areas, and found that there are differences in the effects of traffic accident factors in different urban functional areas, and that the fitting effect of the regional model is better than that of the global model. However, the functional zones are not always the biggest factor of the sub-stratum difference. For example, in this study in Shenzhen, the sub-stratum differences of primary cause and types of primary responsible party were all more obvious than those of zones.

### 6.2. The Influence of Traffic Violation on the Severity of Traffic Accidents

The results of factor analysis show that there are significant differences in the influencing factors of fatalities and injuries, which is basically consistent with the previous research findings [40]. Whether for fatalities or injuries, the primary cause is the most important factor. The casualties caused by traffic violations are significantly higher than those caused by non-illegal reasons [41,42]. Different illegal behaviors and different illegal subjects have different effects on the severity of traffic accidents. The interaction of the two factors further reveals the high-risk behaviors of different subjects.

The role of alcohol as a major factor leading to traffic accidents has been firmly established. The thresholds of driving under the influence of alcohol and drunk driving in mainland China are 0.02 and 0.08, respectively. This study finds that the average fatalities of drunk driving is significantly higher than that of driving under the influence of alcohol, but there is no significant difference in injuries, which implies that lower blood alcohol concentration (BAC) is more effective to reduce fatalities than injuries. This is similar to the previous research conclusion [43,44]. The results of factor interaction further show that drunk driving of large and medium buses is not only the top 10 of high-risk lethal behaviors, but also the top 10 of high-risk injury behaviors, and that drunk driving of heavy trucks and minibuses are all included in the top 10 high-risk lethal behaviors.

Speeding is another widely recognized major cause of traffic accidents. A large number of studies have confirmed that speed can not only affect the frequency of traffic accidents, but also the severity of accidents [45,46]. This study finds that the larger the speeding range is, the more fatalities will be caused, but reducing the speeding range does not significantly reduce injuries. The results of factors’ interaction further show that speeding occupies four seats in the top 10 high-risk lethal behaviors. They are: speeding over 50% for motorcycles, minibuses, large and medium buses, and speeding below 50% for heavy trucks, and in the top 10 high-risk injury behaviors, speeding occupied three seats, which are: speeding below 50% for large and medium buses, speeding over 50% for minibuses, and speeding below 50% for minibuses, respectively.

The results of the sub-stratum comparison of the influencing factors show that overloading is more likely to cause greater casualties than drunk driving. The results of factor interaction further show that heavy truck overloading is one of the high-risk lethal behaviors. Chang and Mannering [47] found that accidents involving trucks are more likely to have serious consequences than non-truck-involved accidents. China’s statistics show that 69.7% of major truck traffic accidents are caused by overloading. The larger the overloading is, the more likely it is to cause major casualties [48]. During 2000–2018 in China, more than 50 bridges collapsed due to overloading of heavy trucks [49]. According to China’s relevant laws, drunk driving is a “crime of dangerous driving”, which will receive criminal punishment, while for the more dangerous overloading of trucks, only administrative penalties such as fines can be imposed. This is one of the main reasons why overloading trucks continue to boom in China. Therefore, it is necessary to promote overloading into the penalty and give full play to the deterrent role of the law to control truck overloading.

There is no rigid barrier to protect pedestrians, cyclists, and motorcyclists, who are usually called vulnerable road users (VRUs) [50]. Previous studies have found that when pedestrians and cyclists are at fault, they are likely to suffer serious injuries, but there are differences, as for motorcycles [51,52,53]. It is found in this study that pedestrian and non-motor vehicles are more likely to cause fatalities than minibuses when they violate the traffic law. On the contrary, motorcycles are more likely to cause injuries than pedestrians, non-motor vehicles, and minibuses. Therefore, the causes of VRUs violations should be given special attention and should be deeply studied.

### 6.3. Influence of Other Factors on Traffic Accidents

#### 6.3.1. Time Factors

The influence of time factors on traffic accidents in the literature is diverse and has mixed effects on road safety due to the regional differences, the different rules of people’s activities, and the different standards of time division. Feng et al. [54] found that autumn and winter increase the probability of more severe accidents, and that day of the week is significantly associated with a great increase in the weekend when compared with weekdays, and that with respect to accidents occurring in the morning, increases in the likelihood of higher severe accidents are observed throughout both evening and night. Pahukula, Hernandez, and Unnikrishnan [55] found that serious injury accidents are less likely to happen in summer, while many serious injury accidents occur between 10:00 a.m. and 3:00 p.m. This study finds that in Shenzhen, the average number of fatalities in winter is significantly higher than that in spring and summer. There is no significant difference in fatalities and injuries between the weekend and weekdays, but the time interval has a significant impact on fatalities and injuries. At the peak hour in the morning and evening, casualties are significantly lower than other times, since speed would be lower in congested situations. While the high incidence of fatalities is from 9:00 to 17:30, when the speed is high and there are many trucks, the high incidence of injuries occurred during 19:30–7:00 the next day, which may be related to the reduction of VRUs which are vulnerable to serious injuries.

#### 6.3.2. Road Factors

This study found that there is high probability of fatalities on highways, elevated sections, and ramps in Shenzhen, similar to previous studies [12]. There are the least fatalities on flat road sections and the most difficult sections (sharp turns and steep slopes). Higher pavement grade and better pavement condition contribute to reduced fatalities. Concrete guardrails are more likely to cause fatalities than other roadside protection facilities.

#### 6.3.3. Environment Condition

The study found that hilly areas are more likely to cause fatalities than plain and mountainous areas, while Dong et al. [24] found rolling and mountainous topography more likely to cause accidents. This paper also found that the situation of no lighting in daytime and at night is more likely to cause fatalities than that of lighting at night, similar to previous research [56].

In view of the above findings, it is necessary to improve the condition of road infrastructure, strengthen the management and control of highways, elevated sections and ramps, and improve the lighting conditions of roads at night, so as to reduce the casualties caused by traffic accidents.

### 6.4. Interaction between Independent Variables

Methods including association rules, decision trees, and hierarchical clustering explore the characteristics of different combinations of factors and enrich the analysis conclusions. Wang et al. [57] applied the boosted regression tree model (BRT) to find that the presence of so many interaction effects indicate that the crash rate at an intersection is dependent on a complex combination of intersection characteristics (not simply additive). Li et al. [58] examined driver injury severity in intersection-related crashes using cluster analysis and hierarchical Bayesian models, and found that a number of crash-level variables, vehicle/driver-level variables, along with some cross-level interactions imposed, significantly influenced driver injury severity. This study further explained the type of interaction among the influencing factors and quantified the intensity of the interaction.

## 7. Conclusions

In this paper, the spatial stratified heterogeneity of traffic accidents in Shenzhen was detected by a geographic detector. The influence factors of fatalities and injuries, the sub-stratum differences of factors, and the types and intensity of the interaction among factors were then analyzed. The main conclusions are as follows: the influencing factors of fatalities and injuries are different, the traffic primary cause and types of primary responsible party have a strong impact on fatalities and injuries, and zones, time interval, and whether illegal have a certain impact on fatalities and injuries, while seasons, multiple road factors, lighting, topography, etc., only have a certain impact on fatalities. There was a nonlinear enhanced relationship between factors. Top 10 high-risk lethal behaviors and high-risk injurious behaviors were found through secondary geographical detection.

The results of this study are helpful for making prevention and control measures. The differences among traffic participants and among traffic violations should be taken into account. Special attention should be paid to heavy truck overloading along with pedestrian and cyclist violations. Furthermore, the results are helpful for the method selection of subsequent research. These methods, including modeling data hierarchically, introducing local variables, or using variable functions will be prioritized.

Some other questions also deserve profound studies. The interaction of factors contains a lot of valuable information. In this paper, what kind of traffic participants will cause greater casualties with what fault has been found out through secondary geographical detection of the combination of primary cause and types of primary responsible party. The second detection of the other two combination factors and the effect of a combination of more than three factors deserve further study. The mechanism behind the phenomenon that the combination of non-significant factors and other factors becomes significant is also worthy of further exploration. Furthermore, this paper only studied the primary cause of traffic accidents, but there is more than one cause in many traffic accidents. The influence of co-occurrence and the interaction of multiple causes on traffic accidents’ severity also deserves further study.

## Figures and Tables

**Figure 1 ijerph-17-00572-f001:**
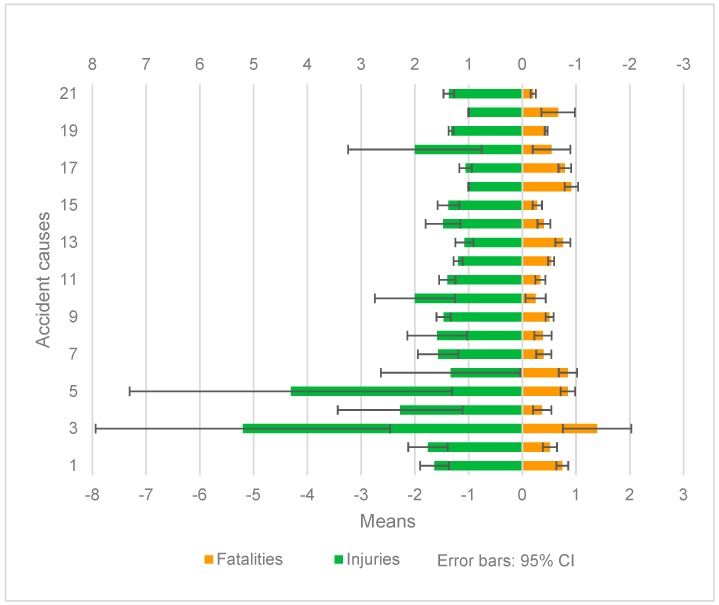
Sub-strata comparison of primary cause.

**Figure 2 ijerph-17-00572-f002:**
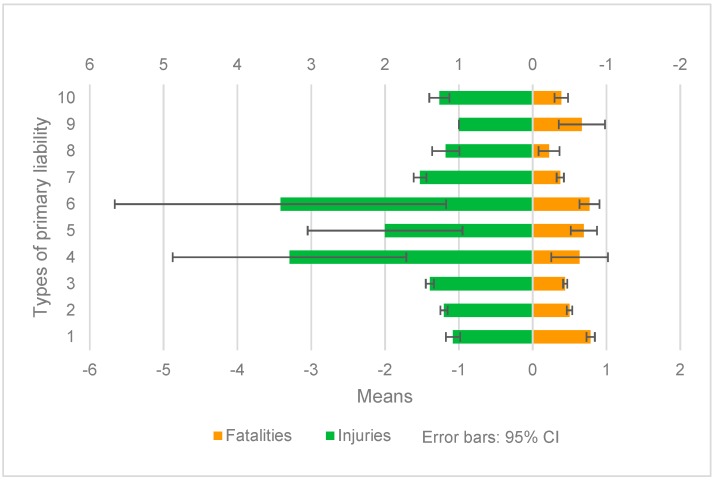
Sub-strata comparison of types of primary responsible party.

**Table 1 ijerph-17-00572-t001:** Definitions and descriptive statistics.

Variables	Definition
Injury Severity	
Fatalities	Number of fatalities in a traffic accident, integer type
Injuries	Number of injuries in a traffic accident without fatality, integer type
Geographical Region	
Zones	1 = City Center zone; 2 = Western coastal zone; 3 = Midland zone; 4 = Eastern zone; 5 = Eastern coastal zone
Time of Occurrence	
Seasons	1 = Spring (March–May); 2 = Summer (June–September); 3 = Autumn (October–November); 4 = Winter (December–February)
Day of the week	1 = Monday; 2 = Tuesday; 3 = Wednesday; 4 = Thursday; 5 = Friday; 6 = Saturday; 7 = Sunday
Time interval	1 = 00:00–06:59 (midnight to dawn); 2 = 07:00–08:59 (morning rush hours); 3 = 09:00–11:59 (morning working hours); 4 = 12:00–17:29 (afternoon working hours); 5 = 17:30–19:29 (afternoon rushing hours); 6 = 19:30–23:59 (nighttime)
Road Factors	
Road type	1 = Highway; 2 = Urban Expressway; 3 = First-class highway; 4 = Second-class highway; 5 = Third-class highway; 6 = Fourth-class highway; 7 = Substandard road; 8 = Branch urban road; 9 = Road in public parking; 10 = Road in public square; 11 = Road in community; 12 = Other road
Road line style	1 = Straight; 2 = General curve; 3 = General slope; 4 = General curve and general slope; 5 = Steep slope; 6 = Sharp curve; 7 = General curve and steep slope; 8 = General slope and sharp curve; 9 = Sharp curve and steep slope
Road section type	1 = Ordinary section; 2 = Plane intersection; 3 = Bridge; 4 = Access; 5 = Internal section; 6 = Elevated section; 7 = Ramp; 8 = Tunnel; 9 = Narrow section
Pavement material	1 = Asphalt concrete; 2 = Cement concrete; 3 = Sand; 4 = Soil; 5 = Others
Pavement condition	1 = Good; 2 = Under construction; 3 = Convex–concave; 4 = Others
Roadside protection	1 = Green belt; 2 = Border tree; 3 = Concrete guardrail; 4 = Protective Pier (column); 5 = Metal guardrail; 6 = Corrugated beam guardrail; 7 = No protection
Management Status	
Traffic sign	0 = Bad or no; 1 = Good;
Lighting condition	1 = Daytime; 2 = Street lighting at night; 3 = No street lighting at night
Environment Condition	
Weather	1 = Sunny; 2 = Cloudy; 3 = Rainy; 4 = Others
Topography	1 = Plain; 2 = Hill; 3 = Mountain
Traffic Violations	
Primary cause	1 = Drunk driving; 2 = Driving under the influence of alcohol; 3 = Speeding over 50%; 4 = Speeding below 50%; 5 = Overloading; 6 = Backing and wrong-way driving on highway; 7 = License violation; 8 = Illegal overtaking; 9 = Traffic signal violation; 10 = Traffic sign violation; 11 = Wrong-way driving, not on highway; 12 = Illegal road occupying; 13 = Illegal backing; 14 = Failure to give way properly; 15 = Illegal meeting; 16 = Helmet violation; 17 = Illegal entering onto highway; 18 = Vehicle defect; 19 = Other violations; 20 = Road facilities hazard; 21 = Other non-illegal fault
Whether illegal	1 = Illegal fault; 0 = Legal fault
Types of primary responsible party	1 = Pedestrians; 2 = Non-motorized vehicles; 3 = Minibuses; 4 = Large and medium buses; 5 = Light trucks; 6 = Heavy trucks; 7 = Motorcycles; 8 = Other motor vehicles; 9 = Traffic management authority; 10 = Others

**Table 2 ijerph-17-00572-t002:** Types and strengths of interaction between two independent variables.

Types and Strengths of Interaction	Discriminant Basis
Weaken, nonlinear	q(X1∩X2) < min(q(X1),q(X2))
Weaken, nonlinear, single	min(q(X1),q(X2)) < q(X1∩X2) < max(q(X1),q(X2))
Enhance, bi	q(X1∩X2) > max(q(X1),q(X2))
Independent	q(X1∩X2) = q(X1)+q(X2)
Enhance, nonlinear	q(X1∩X2) > q(X1)+q(X2)

**Table 3 ijerph-17-00572-t003:** Spatial stratified heterogeneity and contribution of each factor.

Factors	Fatalities		Injuries	
	q Value	p Value	q Value	p Value
Zones	0.011	0.000	0.018	0.000
Seasons	0.003	0.03	-	-
Day of the week	-	-	-	-
Time interval	0.007	0.000	0.008	0.024
Road type	0.037	0.000	-	-
Road line style	0.023	0.000	-	-
Road section type	0.023	0.000	-	-
Pavement material	0.007	0.027	-	-
Pavement condition	0.009	0.002	-	-
Roadside protection type	0.008	0.000	-	-
Traffic sign	-	-	-	-
Lighting condition	0.009	0.000	-	-
Weather	-	-	-	-
Topography	0.016	0.000	-	-
Primary cause	0.094	0.000	0.120	0.000
Whether illegal	0.018	0.000	0.003	0.016
Types of primary responsible party	0.042	0.000	0.097	0.000

“-” indicates that the significance test is more than 0.05, and it is meaningless.

**Table 4 ijerph-17-00572-t004:** Sub-strata comparison of other factors.

Significant Factors of Fatalities or Injuries	Sub-Strata Comparison (Mean Value, 95% Confidence Level)	Interpretation
Zones of fatalities	4 > others	Zone 4 > Other zones
Zones of injuries	4, 5 > 1, 3	Zone 4, 5 > Zone 1, 3
Time interval of fatalities	2, 5 < others 3, 4 > others	Rushing hours < Other time intervals Working hours > Other time intervals
Time interval of injuries	2, 5 < others 1, 6 > others	Rushing hours < Other time intervals Night to dawn > Other time intervals
Seasons of fatalities	4 > 1, 2	Winter > Spring and Summer
Road type of fatalities	1, 5 > 3 11 > 8, 12	Highway and third-class highway > first-class highway and road in community > branch urban road and other road
Road line style of fatalities	1, 9 < others	the straight and sharp curve steep slope is the lowest
Road section type of fatalities	6, 7 > 1, 2, 4	Internal section and elevated section > ordinary section, plane intersection, and access
Pavement material of fatalities	3, 4, 5 > 1, 2	sand, soil, and other pavement > asphalt concrete and cement concrete pavement
Pavement condition of fatalities	3, 4 > 2 > 1	Convex–concave condition and other conditions > under construction condition > good condition
Roadside protection type of fatalities	3 > others	Concrete guardrail is the highest
Topography of fatalities	2 > 1, 3	Hill > plain and mountain
Lighting condition of fatalities	1, 3 > 2	Daytime and no street lighting at night > Street lighting at night

**Table 5 ijerph-17-00572-t005:** Interaction between pairs of factors causing fatalities.

A∩B	Q (A ∩ B)	Q (A + B)	Interaction Type
Primary cause ∩ Types of primary responsible party	0.178	0.135	Enhance, nonlinear
Primary cause ∩ Road section type	0.162	0.117	Enhance, nonlinear
Primary cause ∩ Road type	0.156	0.131	Enhance, nonlinear
Primary cause ∩ Road line style	0.147	0.117	Enhance, nonlinear
Primary cause ∩ Zones	0.136	0.105	Enhance, nonlinear
Primary cause ∩ Roadside protection	0.135	0.101	Enhance, nonlinear
Primary cause ∩ Time interval	0.135	0.101	Enhance, nonlinear
Primary cause ∩ Day of the week	0.129	0.096	Enhance, nonlinear
Primary cause ∩ Topography	0.115	0.110	Enhance, nonlinear
Primary cause ∩ Seasons	0.115	0.097	Enhance, nonlinear

A ∩ B means superposition of A and B; Q (A ∩ B) means the *q* value of A ∩ B; Q (A + B) means the *q* value of A plus the *q* value of B.

**Table 6 ijerph-17-00572-t006:** Interaction between pairs of factors causing injuries.

A ∩ B	Q (A ∩ B)	Q (A + B)	Interaction Type
Primary cause ∩ Seasons	0.421	0.122	Enhance, nonlinear
Primary cause ∩ Zones	0.345	0.138	Enhance, nonlinear
Primary cause ∩ Types of primary responsible party	0.345	0.217	Enhance, nonlinear
Primary cause ∩ Time interval	0.336	0.128	Enhance, nonlinear
Primary cause ∩ Roadside protection	0.319	0.121	Enhance, nonlinear
Primary cause ∩ Day of the week	0.247	0.122	Enhance, nonlinear
Primary cause ∩ Years	0.221	0.122	Enhance, nonlinear
Types of primary responsible party ∩ Day of the week	0.216	0.099	Enhance, nonlinear
Types of primary responsible party ∩ Seasons	0.203	0.099	Enhance, nonlinear
Types of primary responsible party ∩ Topography	0.202	0.099	Enhance, nonlinear

A ∩ B means superposition of A and B; Q (A ∩ B) means the *q* value of A ∩ B; Q (A + B) means the *q* value of A plus the *q* value of B.

**Table 7 ijerph-17-00572-t007:** Top 10 fatalities and injuries of combined factor (primary cause and types of primary responsible party).

Rank	Traffic Violations of Fatalities	Traffic Violations of Injuries
1	Drunk driving of large and medium buses	Speeding below 50% of large and medium buses
2	Illegal entering onto highway of non-motor vehicles	Overloading of heavy trucks
3	Speeding over 50% of motorcycles	Illegal overtaking of large and medium buses
4	Drunk driving of heavy trucks	Drunk driving of large and medium buses
5	Speeding over 50% of minibuses	Speeding over 50% of minibuses
6	Speeding over 50% of large and medium buses	Illegal overtaking of heavy trucks
7	Illegal entering onto highway of pedestrian	Traffic sign violation of light trucks
8	Overloading of heavy trucks	Traffic signal violation of heavy trucks
9	Drunk driving of minibuses	Overloading of light trucks
10	Speeding below 50% of heavy trucks	Speeding below 50% of minibuses

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
