# Peer review of "Geographical Detection of Traffic Accidents Spatial Stratified Heterogeneity and Influence Factors"

_ijerph, 2020, doi:10.3390/ijerph17020572_

Round 1
Reviewer 1 Report
The paper is interesting and well written. However, two important points should be more considered and detailed:
1) In the beginning (Abstract and Introduction), specific research goal should be stated. The research approach may be guided either by hypothesis or it may be exploratory. What was the original research goal and how was it answered?
2) What is the practical application of the findings? How may the be used by practitioners? Or what needs to be pursued further before application is possible?
There are also some occasions of strange phrases, for example "stratified heterogeneous" (should be heteregeneously) or "Noteworthy, there are..." (should be for example Importantly).
Author Response
Thanks for your comments on our paper. We have revised our paper according to your comments.
Please see the attachment.

Reviewer 2 Report
The article deals with an important issue of determining factors which influence the severity of road accidents. The paper is generally well written, but the Authors must justify the adopted methodology (see remark #2).
In the Introduction section – please comment more detailed the weakness of the "independence" assumption. Please justify the assignment of the numbers (numerical values) to particular factors in Table 1. When you for instance assign zones like this “1=City Centre zone; 2=Western coastal zone; 3= Midland zone; 4=Eastern zone; 5=Eastern coastal zone”, you must be aware that they are not ordinary numerical values. This assignment is arbitrary and changing the enumeration order will result in quite different values of mean and standard deviation. Table 1 again, accident causes – actually, it is possible that there are more than one cause. For instance a perpetrator of the accident could be drunken, he/she speeds, ignores traffic signals and illegally overtakes. How to resolve such a situation? Section 5.2.1 (accident causes). You describe the results shown in Figure 1 and you use the numbers introduced in Table 1. This is OK, but could you repeat at least the most important accident causes literally? It would help a reader to understand the results – now he must scroll to Table 1, what is a bit inconvenient. Section 5.2.2 – the same
Author Response

(The authors gave the same response as above.)

Reviewer 3 Report
The paper is very well organized and provides very good view at geographical detection of traffic accidents. The provided datasets and analyses based on the achieved results is very complete and well structured.
However, I believe the paper still lacks a clear description of utilized approach for spatial stratified. I suggest author to elaborate more on this for general readers of journal.
Author Response
Thanks for your comments on our paper. We have revised our paper according to your comments.
Since q-statistics is the basis of Geodetector, we have added a separate section (4.1) to describe its principle. Hoped that these contents can help readers understand Geodetector better.
Round 2
Reviewer 1 Report
Thank you for the revision. I can see that all the review points have been addressed and the paper improved. I recommend to accept it for publication in the journal.
Reviewer 2 Report
Since the Authors have addressed all my remarks I have no further comments